# Crafting a Job among Chinese Employees: The Role of Empowering Leadership and the Links to Work-Related Outcomes

**DOI:** 10.3390/bs14060451

**Published:** 2024-05-27

**Authors:** Mengyan Chen, Yonghong Zhang, Haoyang Xu, Xiting Huang

**Affiliations:** 1Faculty of Psychology, Southwest University, Beibei, Chongqing 400715, China; chenmy2020@email.swu.edu.cn; 2Research Center of Psychology and Social Development, Southwest University, Beibei, Chongqing 400715, China; 3College of State Government, Southwest University, Beibei, Chongqing 400715, China; zhzh@swu.edu.cn (Y.Z.);

**Keywords:** job crafting, empowering leadership, work engagement, in-role performance, Chinese employees

## Abstract

The present study aims to examine the process through which empowering leadership shapes employees’ work engagement and in-role performance by facilitating job-crafting behaviors, specifically seeking resources, seeking challenges, and reducing demands. Based on the extensive data from 733 Chinese employees across various organizations located predominantly in Chongqing and Xi’an, China, we carried out different types of statistical analysis such as confirmatory factor analysis (CFA) and structural equation modeling (SEM) to examine the relationships among empowering leadership, specific job-crafting behaviors, work engagement and in-role performance, test our hypothesis and our conceptual model. The results from structural equation modeling (SEM) suggested that empowering leadership was positively related to employees’ work engagement and in-role performance; empowering leadership was positively related to employees’ job crafting (seeking resources, seeking challenges and reducing demands); seeking resources, seeking challenges and reducing demands were positively related to in-role performance, and seeking challenges and reducing demands were positively related to work engagement. In the relationship between empowering leadership and in-role performance, seeking resources serves as a mediating factor. Similarly, seeking challenges mediates the association between empowering leadership and both work engagement and in-role performance. Furthermore, reducing demands mediates the links between empowering leadership and both work engagement and in-role performance. The implications of these findings are subsequently discussed.

## 1. Introduction

In today’s fast-paced and unpredictable business environment, organizations are increasingly fostering job crafting among employees to enhance their overall performance. Unlike the top-down strategies used by business, job crafting is a self-initiated bottom-up process that enables individuals to customize their responsibilities and roles to fit their skills and needs. From a perspective of Job Demands—Resources (JD-R), job crafting enables employees to change the levels of either job demands or job resources to fit them with their abilities and needs [1]. Aligned with this framework, job crafting emerges as a distinct form of proactive employee behavior, encompassing activities such as seeking resources, seeking challenges, and reducing demands [2]. Existing research has explored the impact of job crafting on work performance; however, few have noticed the different roles of various job-crafting behaviors [1,3]. Indeed, different aspects of job crafting may play different roles in affecting employees’ performance. For example, drawing from the challenge–hindrance stressor framework, job demands can be categorized into two distinct types: challenge demands and hindrance demands [4]. Hindrance demands tend to exhibit negative correlations with job satisfaction and organizational commitment, while positively correlating with turnover intention. Conversely, challenge demands display an opposing pattern of relationships. Seeking challenges may involve employees proactively seeking additional responsibilities and exploring new tasks at work once their current ones are completed. Note that when jobs are far too demanding, demands are less likely to be seen as challenges, and seeking challenges is not an option to consider. In such a case, reducing demands may be viewed as a necessary health-protecting coping mechanism [2].

Additionally, the scarcity of research on contextual factors related to job crafting among employees in organizations significantly hinders our ability to provide organizations with clear guidelines or identify strategies that can effectively enhance employees’ job crafting. Leadership in the organization is considered as a crucial contextual factor based on the existing research [5]. Empowering leadership stands apart from other leadership styles, as it involves the process where leaders delegate power to their employees. From a socio-structural perspective, empowering leadership is a unique set of leader behaviors including leading by example, participative decision-making, coaching, informing and showing individual concern [6]. Within this leadership style, leaders cultivate a profound sense of self-determination, trust, goal-oriented focus, self-confidence, and development support among their subordinates [7]. They foster an environment that encourages employees to take ownership of their leadership and develop self-leadership capacities and skills by optimizing work conditions that enhance employees’ sense of control over significant work goals. Additionally, this leadership approach makes subordinates feel they are entitled and able to pursue their work goals while being responsible for their own decisions [8]. Subordinates are thus given the opportunity to explore solutions to their own challenges and implement the actions independently, without the need for direct supervision [9].

Therefore, the present study aims to explore the role of empowering leadership in crafting a job and its links to work-related outcomes. The specific research questions in this study are as follows: (1) Whether empowering leadership can facilitate employee job crafting and whether there is a difference among the relationships of empowering leadership with different job-crafting behaviors in the context of Chinese organizations? (2) Do different job-crafting behaviors encourage employee work-related outcomes (work engagement and in-role performance) in different ways? (3) Does empowering leadership encourage employee work-related outcomes via employee job crafting? And (4) do different job-crafting behaviors all matter in the relationships between empowering leadership and work-related outcomes? We build on the literature on empowering leadership and employee job crafting to propose that empowering leadership is a key driver of employee job crafting and the empowering leadership-work-related outcomes exist, at least partially, through the underlying process of employee job crafting.

## 2. Literature Review

### 2.1. The Relationship between Empowering Leadership and Work Engagement and In-Role Performance

A large body of research suggested that empowering leadership can be positive for employees and organizations, as its behaviors facilitate employees’ intrinsic motivation and the resulting beneficial outcomes such as work engagement, job satisfaction, creativity, extra-role behaviors and work performance [10,11,12]. Empowering leadership may create supportive and trusting environments in which leaders show respect for their subordinates [13]. Empowering leadership can be beneficial for facilitating employee performance because empowering leader behaviors lead to the satisfaction of the need for autonomy and competence: based on self-determination theory, the satisfaction of these needs stimulates employees to show higher levels of motivation at work and the resulting higher levels of job performance [14]. Amoroso suggested that empowered employees can direct their own emploitation and exploration activities and develop their ability to identify business opportunities as they feel more responsible and competent [15]. Empowering leadership is reported to affect employee work attitudes, behavior and performance either directly or indirectly.

 **Hypothesis 1.** *Empowering leadership is positively related to work engagement (H1a) and in-role performance (H1b)*.

### 2.2. The Relationships between Empowering Leadership and Job Crafting

#### 2.2.1. The Relationship between Empowering Leadership and Seeking Resources

Empowering leadership can be effective for generating resources in some forms, such as decision latitude and leader support. Perceived job resources as well as psychological resources are important drivers of proactive behaviors. Kim et al. argued that employees working in a resourceful environment characterized by autonomy, learning opportunities, and high-quality coaching tend to have greater confidence in their abilities and are better able to customize their work environment to align with their personal needs and strengths [9]. As a global factor, empowering leadership functions as an occupational resource that can enhance employees’ perceived ability to cope with job demands [16]. This, in turn, leads to positive outcomes both at the individual and organizational level, such as improved well-being, a sense of meaningfulness, creativity and overall performance.

 **Hypothesis 2a.** *Empowering leadership is positively related to subordinates’ seeking resources*.

#### 2.2.2. The Relationship between Empowering Leadership and Seeking Challenges

Under empowering leadership, employees experience a heightened feeling of freedom or discretion, which facilitates their opportunity perception of what they can do in their jobs and how they do it. Furthermore, perceived opportunities for job crafting tend to be psychologically positive, because these opportunities suggest autonomy to act, a sense of possible gain and some perceptions of capability or means to act [17]. Thus, employees having such an opportunity perception are more likely to discover the paths available in how they enact their jobs and in turn show more actual job-crafting behaviors to enhance the engagement, meaning and satisfaction associated with their work. The characteristics of empowering leadership could stimulate employees’ different stress categories in various ways, specifically increasing challenge stress but decreasing hindrance stress [6].

 **Hypothesis 2b.** *Empowering leadership is positively related to subordinates’ seeking challenges*.

#### 2.2.3. The Relationship between Empowering Leadership and Reducing Demands

In addition, the views of empowering leaders regarding the tasks and workloads assigned to employees affect employees’ demand-reducing behaviors, probably because through delegating more power, their subordinates may feel that they have the potential to relieve themselves of some of the tedious and repetitive tasks and eventually have more time and freedom to do what they like and value [18]. Furthermore, empowering leadership might sometimes involve high levels of job demands such as work pressure and role ambiguity. In such a case, it is likely that employees might employ self-protective strategies such as simplifying the complexity of the tasks at work or avoiding making difficult decisions at work so as to prevent their personal resources and energy from further depletion. According to empowering theory, hindrance stressors are seen as roadblocks to goals that subordinates cannot circumvent and work at cross-purposes to empowering leadership. Employees could cut corners via unethical means as a fallback to reduce such stressors because playing by the rules, requiring succumbing to an organizational context characterized by hindrance stressors, is likely perceived as constraining or futile in achieving desired outcomes [19]. Therefore, subordinates of empowering leaders may engage in demand-reducing behavior.

 **Hypothesis 2c.** *Empowering leadership is positively related to subordinates’ reducing demands*.

### 2.3. The Relationships among Job Crafting and Work Engagement and In-Role Performance

Expanding upon the JD-R framework, job crafting may improve employees’ person-job fit by creating a balance between job needs and job resources. This, in turn, may lead to increased job satisfaction and favorable organizational outcomes. Although job crafting is important and is thought to be associated with work-related outcomes, different theoretical perspectives have led to differing findings [1]. Previous studies have suggested that increasing resources and challenges is linked to important work outcomes such as job satisfaction, work engagement [20], meaningfulness [21], and improved job performance [22]. According to the Regulatory Focus Theory, there are two distinct motivational systems that each have their own set of basic demands. The approach-oriented strategy is the first motivational system, in which people employ high-commitment feelings as their preferred emotional methods and eagerness as their main approach for pursuing their goals. Achieving goals and objectives is the primary focus of those who are promotion-oriented. The second motivational system is avoidance-oriented, in which people choose low-commitment feelings as their preferred emotional means and vigilance as their chosen approach for achieving their goals: Prevention-oriented people show strong care about upholding their duties, commitments, and responsibilities in order to avoid failures [23]. Therefore, seeking resources and seeking challenges are generally more approach-oriented, where employees actively engage with and directly address the problems they encounter. Reducing demands differs from the other two job-crafting behaviors in a crucial aspect. It tends to be an avoidance-oriented strategy, allowing individuals to circumvent workplace issues.

#### 2.3.1. The Relationships among Seeking Resources and Work Engagement and In-Role Performance

The linkage between seeking resources, work engagement, and in-role performance can be elucidated through the Job Demands-Resources (JD-R) framework. Job resources are defined as “those physical, psychological, social or organizational aspects of the job that may do any of the following: (a) be functional in achieving work goals; (b) reduce job demands and the associated physiological and psychological costs; (c) stimulate personal growth and development” [24]. Specifically, by expanding job resources (e.g., skill variety and autonomy), employees may perceive more personal control over their jobs and feel more competent and autonomous when there are increased levels of skill variety, autonomy, and learning opportunities [1]. Increasing resources stimulates employees’ internal motivation and external motivation internalization, making them more willing to devote themselves to work, and the work resources they gain can provide direct support for them to complete tasks and achieve goals. For example, when employees encounter problems in work, by asking colleagues or superiors for help, they can not only feel the support from leaders and colleagues, but also obtain the information, resources and methods they need to solve the problem, thereby improving work quality and achieving good performance.

 **Hypothesis 3.** *Seeking resources is positively related to employee work engagement (H3a) and in-role performance (H3b)*.

#### 2.3.2. The Relationships among Seeking Challenges and Work Engagement and In-Role Performance

Li Zong-bo and Li Rui (2013) [25] pointed out that increasing challenging job demands, such as assuming more responsibilities after completing assigned tasks or seeking more challenges, may make individuals feel stressed to some extent, but it can also keep them motivated, prevent boredom, and enable them to gain benefits and rewards in terms of knowledge, skills, and personal growth after successfully coping with challenges, thereby generating a strong sense of accomplishment. Challenging job demands enable employees to fully utilize their knowledge and skills, perceive work challenges as important ways for self-development and improvement, generate more positive emotions, have a positive work attitude, proactively solve various problems in the workplace, and achieve higher levels of work performance [26]. According to cognitive interaction theory, challenging job demands can bring potential benefits and growth to individuals, thereby stimulating their internal motivation, evoking positive emotions, and prompting individuals to adopt proactive or problem-solving-oriented coping strategies, such as increasing their effort levels and actively trying to find solutions to problems. Additionally, based on expectancy theory, when individuals believe that challenging job demands will bring them benefits and rewards, they tend to actively undertake challenging work tasks and become more proactive in their work efforts [27].

 **Hypothesis 4.** *Seeking challenges is positively related to employee work engagement (H4a) and in-role performance (H4b)*.

#### 2.3.3. The Relationships among Reducing Demands and Work Engagement and In-Role Performance

Regarding the relationship among reducing demands and work engagement and in-role performance, previous research has indicated that reducing demands aims to reduce psychological, emotional, or physical demands at work, representing an avoidance-oriented coping strategy that is prone to inducing adverse work experiences (such as burnout and job dissatisfaction) and negative work behaviors (such as procrastination and absenteeism), which are not conducive to employees’ work engagement and performance [2]. Over time, there is a reciprocal relationship between reducing demands and exhaustion. This implies a vicious cycle whereby exhausted employees reduce demands, which increases their burden and intensifies their exhaustion. Employees who avoid demand-crafting also do not perform as well since they do not fulfill the role requirements [3]. Given that, we believe that as employees exhibit more demand-reducing behaviors, they are less likely to maintain high levels of engagement in their work and attain satisfactory performance outcomes.

 **Hypothesis 5.** *Reducing demands is negatively related to employee work engagement (H5a) and in-role performance (H5b)*.

### 2.4. The Mediating Role of Employee Job-Crafting Behaviors

Job crafting emphasizes the process of employees actively adjusting and improving their working environment. When employees perceive the empowering behavior of their leaders, they will, on the one hand, be motivated by more autonomy and show more positive working behaviors, such as taking the initiative to contact other people in the workplace, expanding their own relational resources, and taking the initiative to learn new content to improve their own abilities, but on the other hand, they may adopt avoidance-oriented coping strategies, such as task avoidance, because of the increased psychological burden and excessive consumption of their own resources due to more job responsibilities and work pressure, which may ultimately affect their work engagement [28]. The mediating role of seeking resources and seeking challenges specifically can be explained in terms of the work resources and individual–situational interaction perspectives. From the work resource perspective, empowered leaders provide subordinates with a wide range of organizational and work-related resources, which can lead to individuals’ intrinsic motivation to make positive adjustments in the areas of work tasks and relationships, and thus enhance the fit between themselves and their work, thereby improving their performance and well-being. From the perspective of personal–situational interaction, empowering leadership as an organizational contextual factor can create a free, flexible and safe working environment for employees, making them feel trusted and recognized, which creates an optimal working atmosphere for employees to put their energy and resources into their work roles, and then prompts them to take the initiative in exploring strategies to carry out their work tasks, improve their interpersonal relationships, and show more performance. This in turn encourages employees to explore strategies to carry out their tasks, improve their interpersonal relationships, and exhibit more-extended job reinvention behaviors (i.e., increased resources and challenges), leading to high levels of work engagement and performance.

Finally, the possible mediating role of reducing demands in the influence of empowering leaders on work engagement and performance can be explained by the “double-edged sword” effect of empowering leaders. In other words, empowerment is a “double-edged sword” that enhances the motivation of employees while at the same time potentially increasing the motivation of certain individuals. In other words, authorization is a “double-edged sword” that can increase the work motivation of employees, while at the same time, it may also increase the work responsibilities and work pressure of some employees, which in turn may have a negative impact on their physical and mental health and work status [29]. Employees who perceive leadership empowerment need to take on more job responsibilities and endure more psychological, emotional and physical demands, and in such a situation, their own energy may be exhausted and their own resources may be overly consumed. According to the theory of resource conservation, when faced with a loss of resources, individuals usually take action and reduce resource consumption first to prevent the continued loss of resources and avoid falling into a loss spiral [30]. Therefore, we believe that empowerment from leaders may also cause employees to exhibit requirement-reducing behaviors, such as task avoidance and procrastination, which are detrimental to their work engagement and achievement of work goals.

Based on the abovementioned, we propose:

 **Hypothesis 6.** *Seeking resources plays a mediating role in the relationship between empowering leadership and work engagement (H6a), and also in the relationship between empowering leadership and in-role performance (H6b)*.

 **Hypothesis 7.** *Seeking challenges plays a mediating role in the relationship between empowering leadership and work engagement (H7a) and also in the relationship between empowering leadership and in-role performance (H7b)*.

 **Hypothesis 8.** *Reducing demands plays a mediating role in the relationship between empowering leadership and work engagement (H8a) and also in the relationship between empowering leadership and in-role performance (H8b)*.

 **Hypothesis 9.** *After mediation by employee job crafting, empowering leadership is positively related to work engagement (H9a) and in-role performance (H9b)*.

The conceptual model is depicted in Figure 1. 

## 3. Methods

### 3.1. Participants and Procedures

The current study was approved by the Ethics Committee of the Faculty of Psychology, Southwest University. The participants were recruited by two voluntary research assistants and by the authors of this paper. To ensure the quality of the data, the research assistants were provided with detailed information about the objectives and procedures of the current study as well as the difficulty of the data collection. Participants were recruited through various channels, including direct contact with human resources departments across different companies and personal networks of the research assistants and authors. This approach likely contributed to the diversity of the sample and enhanced the generalizability of the research findings. In total, 867 Chinese employees participated in the survey study, of which 361 employees filled out the online questionnaires, and 506 employees filled out the paper-and-pencil questionnaires. The survey encompassed demographic variables and was accompanied by a cover letter that outlined the study’s objectives and reassured respondents of their confidentiality and anonymity. Because we were interested in the effects of demographics (e.g., age, gender, tenure, education, position) on our study variables, we kindly requested our participants to fill out all demographics.

Accordingly, we removed 134 participants who failed to fill out their demographics from further analyses, resulting in a final sample of 733 Chinese employees dispersed across a wide range of industries such as education, health services, financial services, wholesale and retail, as well as information technology. Of the respondents, 39.4% were females and 60.6% were males, 55.8% had a university-level education, and 33% had a leadership position. Moreover, 32.1% were aged under 25 years, 26.9% between 26 and 30 years, 11.2% between 31 and 35 years, 11.1% between 36 and 40 years, and 18.8% above 41 years. Among the respondents, 25.5% had worked in their current organization for a period of less than 1 year, 31% between 1 and 3 years, 16.5% between 4 and 6 years, 8.6% between 7 and 10 years, and 18.4% for more than 10 years.

To mitigate the potential for common method variance, we took several pre-control measures. Firstly, to ensure the anonymity and confidentiality of all survey responses, we employed site coding as a means of tracking the data, rather than utilizing respondent names. Furthermore, surveys were returned directly to the researchers, further protecting the privacy of the participants. Additionally, we conducted Harman’s single-factor test to assess the presence of common method variance among empowering leadership, job crafting and work-related outcomes. The results of the test indicated that the first factor only explained 18.118% of the variance, falling short of the 40% threshold. This suggests that a single factor did not account for a significant portion of the variation, thereby indicating that the issue of common method variance was not substantial.

### 3.2. Measures

#### 3.2.1. Job Crafting

Job crafting was measured by using a modified 12-item scale by Tims et al. (2012) [31] and Petrou et al. (2012) [2]. The three dimensions—seeking resources, seeking challenges, and reducing demands—have been retained and labeled accordingly. For the purposes of this study, items were chosen based on preliminary factor loadings and were specifically tailored to reflect the work environment of Chinese employees. These items were rated using a 5-point scale, ranging from 1 (totally disagree) to 5 (totally agree). The dimension of seeking resources encompassed five items, such as “I strive to acquire new knowledge and skills at work”, achieving a Cronbach’s alpha reliability coefficient of 0.82. The dimension of seeking challenges comprised four items, including “When an intriguing project arises, I proactively offer to be a part of it as a project collaborator”, with a Cronbach’s alpha of 0.76. Lastly, the dimension of reducing demands consisted of three items, for instance, “I make efforts to ensure that my work is less emotionally draining”, achieving a Cronbach’s alpha of 0.64.

Because the set of 12 items tapped different aspects of job crafting, we carried out confirmatory factor analysis to identify whether all of the job crafting items were loaded onto their respective factors. The overall scale yielded a coefficient alpha of 0.84. The scale validity was acceptable, with $\chi^{2}$/df = 3.20, RMSEA = 0.06, CFI = 0.94, IFI = 0.92, and SRMR = 0.05. The seeking resources factor was measured using five items with factor loadings ranging from 0.646 to 0.733 and an AVE of 0.480. The seeking challenges factor was measured using four items, with factor loadings ranging from 0.519 to 0.719 and an AVE of 0.450. Similarly, the reducing demands factor was assessed through three items, exhibiting factor loadings between 0.586 and 0.627 and an AVE of 0.375.

#### 3.2.2. Empowering Leadership

Employees’ perceptions of empowering leadership were evaluated with a shortened version of the 24-item empowering leadership behavior developed by Wang et al. (2008) [32]. Before the formal survey was conducted, we interviewed 11 supervisors and employees in enterprise to discuss the characteristics of empowering leadership. Additionally, through factor analysis, 14 items were selected, as these items could best represent the delegation of authority (e.g., “my supervisor gives me responsibilities”, factor loading range from 0.771 to 0.811, AVE = 0.626), participation in decision-making (e.g., “when coming to a decision, my supervisor encourages me to express ideas and opinions”, factor loading range from 0.708 to 0.826, AVE = 0.592), coaching (e.g., “my supervisor helps me out when I come across the difficulties”, factor loading range from 0.778 to 0.808, AVE = 0.624) and informing (e.g., “my supervisor explains the company’s rules and expectations to me”, factor loading range from 0.683 to 0.778, AVE = 0.529) in the Chinese context. Items were rated on a 5-point scale, with 1 indicating totally disagreement and 5 indicating totally agreement. The scale yielded a coefficient alpha of 0.93. The scale validity was acceptable, with $\chi^{2}$/df = 4.13, RMSEA = 0.07, CFI = 0.96, IFI = 0.95, SRMR = 0.03.

#### 3.2.3. Work Engagement

The Utrecht Work Engagement Scale [33], consisting of nine items organized into three subscales of three items each, was utilized to assess work engagement. One of the subscales, Vigor, encompassed statements such as “At my work, I feel bursting with energy”, Cronbach’α = 0.87, factor loading range from 0.746 to 0.898, AVE = 0.705; Dedication encompassed statements such as “I am enthusiastic about my job”, Cronbach’α = 0.88, factor loading range from 0.779 to 0.892, AVE = 0.706, and Absorption encompassed statements such as “I am immersed in my work”, Cronbach’α = 0.69, factor loading range from 0.709 to 0.746, AVE = 0.530. Items were rated on a 5-point scale, with 1 indicating total disagreement and 5 indicating total agreement. The scale yielded a coefficient alpha of 0.92.

#### 3.2.4. In-Role Performance

To assess in-role performance, we adopted five items derived from Williams and Anderson (1991) [34]. An illustrative item reads, “I effectively fulfill my assigned duties” (Cronbach’s α = 0.87). Respondents rated each item on a 5-point scale, ranging from 1 (totally disagree) to 5 (totally agree). The factor loading ranged from 0.700 to 0.847, AVE = 0.581.

#### 3.2.5. Control Variables

Age, gender, education, tenure and position were applied as control variables in the analysis. In the existing research, work behavior can be found to be influenced by demographic variables such as age, gender, education, tenure and position. For example, Berg et al. (2010) [35] suggested that employees’ ranks at work are related to the characteristics of job challenges at work and the way they cope with these challenges. Therefore, we applied these variables as control variables.

## 4. Results

Table 1 presents descriptive statistics along with zero-order correlations for the key variables. Collectively, these zero-order correlations show support to the proposed model.

### 4.1. Examining the Discriminability of Study Variables

Prior to verifying the research hypotheses, confirmatory factor analysis (CFA) was conducted to assess the discriminability of the study variables. As shown in Table 2, the results of CFA indicated that the fit between the observed data and the six-factor model was excellent, while the alternative models exhibited poorer fit with the observed data, indicating good discriminant validity of the study variables.

### 4.2. Examining the Direct Effect of Empowering Leadership on Work Engagement and In-Role Performance

The direct effect of empowering leadership on work engagement and in-role performance was firstly examined through structural equation modeling (SEM). The results showed that, after controlling the demographic variables, the model of the direct effect analysis provided a good fit to the data,$\chi^{2}$/df = 3.3411, CFI = 0.961, TLI = 0.946, and RMSEA (90% CI) = 0.057 (0.048, 0.065). Empowering leadership positively predicted employees’ work engagement (β=0.567, *p* < 0.01) and in-role performance (*β*
= 0.392, *p* < 0.01). Therefore, H1a and H1b were supported.

### 4.3. Analysis of the Role of Job Crafting in the Relationship between Empowering Leadership and Work-Related Outcomes

We tested the hypothesized structural model using Mplus 6.0. Multiple indices of fit were calculated to assess the models, such as $\chi^{2}$/df, CFI, TLI, AIC, BIC, adjusted BIC, and RMSEA (90% CI). A high chi-square value suggests that the model does not fit the data appropriately, and a chi-square ratio of three or below is generally considered a suitable criterion for accepting a model. Given that the present data were not multivariate normal, ML estimation may not have been suitable, and the typical chi-square difference test relying on the goodness-of-fit test statistics of individual models did not follow an $\chi^{2}$ distribution. We performed the scaling corrections using MLM estimation to improve the chi-square approximation. The hypothesized model provided a good fit to the data, $\chi^{2}$/df = 5.005, CFI = 0.940, TLI = 0.92, and RMSEA (90% CI) = 0.07 (0.07, 0.08). We conducted additional analyses and compared the hypothesized model against different rival models (see Table 3). Specifically, we first compared the hypothesized mediated model with a rival model (Model 1) removing the direct path from empowering leadership to in-role performance. This model was a slight improvement in the fit, and the AIC, BIC, and adjusted BIC for this model were lower than the hypothesized model. Second, we compared the hypothesized model with a model removing the path from empowering leadership to work engagement. The data did not align well with this model, indicating a significantly poorer fit compared to the hypothesized model. This underscores the significance of the direct link between empowering leadership and work engagement. Additionally, when we contrasted the hypothesized model with a fully mediated model, it also exhibited a poor fit to the data, significantly inferior to the hypothesized model. This further emphasizes the cruciality of the direct pathways. In summary, the hypothesized model was enhanced by eliminating the direct path connecting empowering leadership to in-role performance. The final model was a good fit, $\chi^{2}$/df = 4.452, CFI = 0.942, TLI = 0.901, and RMSEA (90% CI) = 0.069 (0.061, 0.076).

Figure 2 shows the significant pathways for the final model. Providing support for H2, namely H2a, H2b and H2c, empowering leadership was positively related to seeking resources (*β* = 0.583, *p* < 0.001), seeking challenges (*β* = 0.515, *p* < 0.001) and reducing demands (*β* = 0.408, *p* < 0.001). Contrary to H3a and H5a, seeking resources did not have a significant unique association with work engagement, whereas reducing demands did positively relate to work engagement (*β* = 0.176, *p* < 0.001). Supporting H4a, seeking challenges was positively related to work engagement (*β* = 0.333, *p* < 0.001).

In relation to H3b, H4b, and H5b, all three job-crafting behaviors—seeking resources, seeking challenges, and reducing demands—exhibited significant and positive associations with in-role performance, with β values of 0.290, 0.247, and 0.246, respectively, and all at a significance level of *p* < 0.001. Furthermore, regarding H7a and H8a, empowering leadership positively influenced work engagement through its impact on seeking challenges (β = 0.172, *p* < 0.001) and reducing demands (β = 0.072, *p* < 0.001). It is noteworthy that, interestingly, empowering leadership did not demonstrate a significant relationship with work engagement via seeking resources (β = 0.033, *p* > 0.05). However, given that seeking resources was not significantly associated with work engagement, this finding is less remarkable. Finally, in relation to H6b, H7b, and H8b, empowering leadership positively related to in-role performance through its effects on seeking resources (β = 0.169, *p* < 0.001), seeking challenges (β = 0.127, *p* < 0.001), and reducing demands (β = 0.100, *p* < 0.001), further establishing its comprehensive impact on job performance outcomes.

Overall, with the exception of the relationships between seeking resources and reducing demands and work engagement, other direct and indirect pathways were confirmed. Empowering leadership exhibited a positive correlation with seeking resources, seeking challenges, and reducing demands. These three job-crafting behaviors were, in turn, positively associated with in-role performance. Additionally, seeking challenges and reducing demands displayed a positive relationship with work engagement. Regarding H9a, empowering leadership indirectly influenced work engagement through job-crafting behaviors, specifically seeking challenges and reducing demands, while also exerting a direct positive impact. Nevertheless, in contrast to H9b, empowering leadership’s positive relationship with in-role performance was fully mediated by job-crafting behaviors, encompassing seeking resources, seeking challenges, and reducing demands.

In addition to the tests above, consistent with the call for greater attention to the strong impact of job attitudes (e.g., work engagement) in the links between empowering leadership and performance outcomes, we conducted an additional mediation analysis to investigate whether the relationship between empowering leadership and in-role performance would also be mediated by work engagement. More specifically, to test the mediating role of work engagement in the links between empowering leadership and in-role performance, we formulated a structural model, adding the path from work engagement to in-role performance based on the hypothesized model. The results showed that including this path did not significantly improve the fit indices for the model (Δ=1.492, Δdf = 1, *p* > 0.05) and work engagement did play a mediating role in the link between empowering leadership and in-role performance, with the exception of three job-crafting dimensions. In other words, empowering leadership positively linked to job crafting and work engagement, which, in turn, related to in-role performance.

## 5. Discussion

Based on the extensive data from 733 Chinese employees across a variety of organizations located predominantly in Chongqing and Xi’an, China, the results show support for the empowering leadership →employee job crafting →work-related outcomes linkage. In addition, we found that employee work engagement plays the mediating role in the relationships between empowering leadership and in-role performance. However, the data also revealed that seeking resources is not related to work engagement, which is different from the previous studies. This research provides new insights into the role of job crafting and its different dimensions in the relationships between empowering leadership and work-related outcomes.

### 5.1. The Relationship between Empowering Leadership and Job Crafting

The present study found that empowering leadership and distinct job-crafting behaviors are positively correlated. That is, higher levels of empowering leadership can predict different job-crafting strategies. SEM also found that empowering leadership can positively predict job crafting. These findings illustrate that empowering leadership encourages employees to use different job-crafting strategies to adjust their jobs, which is in line with past research [9,36]. Wrzesniewski and Dutton (2001) [17] suggested that empowering leadership behavior can be seen as a necessary premise for job-crafting behavior. Thus employees in work can proactively craft their work and relations. Demerouti (2014) [37] also found that job-crafting behaviors, such as employees’ proactive behaviors in work, can be influenced by work autonomy. The typical features of empowering leadership are delegating power to subordinates and encouraging employees to take ownership of their work, by which employees can perceive more opportunities for job crafting and thus reshape their own jobs and enhance person–job fit. Our findings offer support for the existence of positive relationships between empowering leadership and job crafting.

In addition, these results suggest that empowering leadership affects three job-crafting behaviors differently. Compared to reducing demands, empowering leadership has stronger impacts on seeking resources and seeking challenges. So, when employees perceive a high level of empowering leadership, they would show more proactive behavior to increase job resources and challenges at work. This finding is in line with organizational practice. In our survey, empowering leadership was strongly correlated with employee initiative and proactivity. Several participants noted that when they perceive empowering by their supervisors, they tend to catch up with new ideas to achieve organizational goals and use effective strategies to finish their jobs as well as initiatively communicating with leaders, colleagues or clients to acquire the necessary information and resources for their work. Moreover, the existing studies had similar findings. For example, compared to reducing demands, the relationship between empowering leadership and increasing job resources and job challenges was more closed [36].

### 5.2. The Relationships between Job Crafting and Work Engagement and In-Role Performance

Correlation and SEM analysis suggest that different forms of job crafting (seeking resources, seeking challenges and reduce demands) have positive relations with in-role performance and positively predict in-role performance. By job crafting, employees can achieve higher work performance. Job crafting is defined as employees taking the initiative to change their jobs to achieve a fit between abilities and demands and job demands and job resources. This suggested that through job crafting, employees can create a better work environment to improve person–job fit for achieving optimal functioning at work. To be specific, firstly, by increasing job resources, employees can obtain work-related knowledge and skills, and establish stronger relationships with their coworkers, supervisors and leaders, which can be seen as a basic requirement for great performance. Secondly, by increasing job challenges, employees can engage in more job tasks related to organization goals, and utilize their own knowledge and skills at work to solve problems proactively. This can help them improve their job performance. Finally, by reducing job demands, employees can gain opportunities to disengage from certain burdensome and basic work tasks and focus their personal resources, energy, and time on key role tasks. This approach results in improved in-role performance. In addition, the results showed that compared to seeking challenges and reducing demands, seeking resources is closely related to in-role performance and can be seen as a better predictor of in-role performance. These results are in line with a previous study. For example, Rudolph et al. (2017) [38] found that seeking resources can have a maximum impact on job performance, as well as match the motivational activation pathway hypothesis in the job demands-resources (JD-R) model. Indeed, job resources are those characteristics providing support and help for employees, such as information resources and social support. These resources can be helpful for employees to finish their job tasks and achieve their job ambitions and greater performance.

Additionally, the results suggested that three different job-crafting strategies are positively related to work engagement, and seeking resources and seeking challenges can predict work engagement positively, but seeking resources cannot be a predictor of work engagement. Firstly, seeking challenges can predict work engagement, which is in line with the existing research [20,26]. This suggested employees who proactively seek challenges at work can perceive a higher level of work engagement. Seeking challenges can be a type of demand that can stimulate employees’ intrinsic motivation and sense of achievement, resulting in positive emotions and work outcomes [27]. With increased job challenges, employees are able to engage in more tasks that they show interest in, and they can fully utilize their skills and strengths, which results in positive emotions and job experiences for employees, facilitating their immersion in work tasks and stimulating work engagement [39].

We also found that increasing job resources cannot predict work engagement, which is different from the previous study. However, this result is understandable because Chinese employees influenced by collective cultural values emphasize harmony in daily life; therefore, seeking resources can result in relationship conflicts and task conflicts and thus weaken the positive impacts on work engagement. Yinkui et al. (2016) [26] similarly found that increasing job resources has no effects on work engagement in the Chinese context. Similarly, another study also suggested that increasing job resources has no significant impact on work engagement [40]. This is mainly because seeking resources is involved in the satisfaction of short-term needs, such as improving specific workflows and obtaining assistance for solving specific problems. The benefits generated by these resources may only manifest in helping employees solve current problems without any impact on long-term positive emotions. At the same time, for those employees who possess the ability and motivation to increase work resources, the beneficial effects of obtaining these resources on individual well-being cannot be sustained if there is no opportunity to utilize them. This study suggests that the lack of a significant influence of increased work resources on work engagement may be due to the Chinese organizational context, where increasing job resources typically occurs when employees encounter confusion or problems at work. Although increasing job resources allows employees to address these issues and obtain specific work resources required to complete work tasks, these issues themselves represent adverse work situations. Consequently, the positive effects of increased job resources may be offset by the negative effects of the adverse work experiences, resulting in the lack of association between increased resources and work engagement.

Finally, although the positive relationships between reducing demands and work engagement in this study may differ from previous research, it can be understood from the perspective of conservation of resources (COR) theory. By avoiding work situations with excessive demands, employees can protect their own resources from excessive depletion, mitigate emotional exhaustion, and preserve well-being, which contributes to a renewed cycle of work engagement. Additionally, Demerouti and Peeters (2018) [41] suggested that not all reductions in demands result in low levels of work engagement. The optimization-oriented approach adopted by employees can predict positive work engagement, which to some extent, provides support for the positive relations between reducing demands and work engagement.

### 5.3. The Mediating Role of Job Crafting

The mediation analysis revealed that three distinct types of job-crafting behaviors function as mediators in the association between empowering leadership and in-role performance. More precisely, seeking resources and seeking challenges play a mediating role in the connections between empowering leadership, work engagement, and in-role performance. This is in line with past research [9] that found that increasing structure job resources, increasing social job resources and increasing challenging job demands mediated the relationships between empowering leadership and employee general well-being and subjective career success. The COR theory suggests that possession of certain resources can pave the way for obtaining more resources, creating a spiral effect. Employees often perceive empowering leadership as a valuable resource that prompts them to make self-directed adjustments to their job roles, thereby enhancing other job resources. By proactively shaping their job environments to be both enriching and challenging, individuals are able to achieve a more harmonious match between themselves and their work. It is, therefore, crucial to encourage and provide employees with the freedom to customize their jobs, as job crafting has the potential to inspire enthusiasm, heighten career satisfaction and commitment, and prompt overall well-being [9]. Ji et al. (2023) [1] discovered that enhancing job resources and embracing challenging job demands serve as mediators between leader–member exchange and the flow experienced at work. Furthermore, Khalil et al. (2023) [42] explored the interconnections between servant leadership, job-crafting behaviors, and work outcomes such as engagement and job satisfaction. They observed that both work outcomes positively correlate with seeking resources and challenges, while exhibiting a negative association with reducing demands. Notably, job-crafting behaviors mediate the relationship between servant leadership and these work outcomes. Additionally, the mediation analysis underscores the positive mediating role of reducing demands in the linkage between empowering leadership, work engagement, and in-role performance. This suggests that employees can attain higher levels of work engagement and performance by effectively managing and reducing demands. Indeed, this finding is inconsistent with our hypothesis, but it is also understandable. This is because employees perceived empowering leadership as allowing them to assume more responsibilities and bear more work pressure; on the other hand, employees have more freedom and space to adjust their work characteristics and change their work environment based on their own ideas, which can be helpful to employees in reducing obstacles at work, alleviating work pressure and exhaustion, thus enabling them to perform better in the next round of work [26]. In addition, when leaders empower their subordinates, subordinates have the work autonomy to make decisions about their work. As a result, subordinates are able to disengage from tight schedules, heavy workloads and routine tasks, allowing them to dedicate more time to activities that interest them. This can be positive for promoting their work engagement and performance.

### 5.4. Limitation and Future Research

Despite receiving strong empirical support, our study has several limitations. Firstly, the measures employed in this study rely on self-reported data collected from employees at a single time point. This approach could potentially introduce many different biases (such as common method variance) that call into question its accuracy. In addition, although a number of psychometrically validated questionnaires have been adopted, little attention has been paid to the fact that questionnaire measures do not really measure leader and employee behavior, but rather confuse descriptions of behavior with subjective evaluations of behavior [43]. While we attempted to mitigate this bias through pre-control methods such as ensuring respondent anonymity and providing uniform instructions emphasizing that there are no right or wrong answers, this approach may still have a negative effect on the reliability of the results obtained. Therefore, we strongly suggest future research to explore more robust designs and adopt diverse methods to minimize such biases. For example, future studies could consider incorporating a mixed approach for data collection, combining self-evaluations with peer and leader assessments, and conducting surveys across multiple time points. Tims et al. (2012) [31] suggested that although job crafting is considered as a spontaneous behavior of employees, other individuals in the workplace, such as colleagues, are also capable of noticing the employee’s job-crafting behavior to a large extent and making evaluations that are consistent with the employee’s self-assessment of job crafting.

Secondly, since this was a cross-sectional study, the obtained findings cannot establish causal relationships among the involved variables. Future researchers may adopt a longitudinal design to address any concerns related to causality and make causal statements. In addition, the diary study method and weekly diary design can be adopted to collect data to analyze short-term fluctuations in employees’ behaviors and attitudes, which might reduce the memory bias, thereby enhancing the accuracy of conclusions. Additionally, the sample of the current study involved a wide range of occupational types, job natures and industries, which may raise questions about whether the findings of the present study can be applicable to specific work populations. Future research can pay attention to specific types of occupational populations to investigate such proposed relationships, which would enhance the ecological validity of the research findings.

Finally, the research on ethical leadership has become a promising topic. Although ethical leadership is a different leadership style from the empowering leadership assumed in the current study, it has significant positive correlations. For example, ethical leadership can also enhance employees’ sense of psychological empowerment [44,45]. Ethical leadership focuses on the ethical aspects of work behavior and assumes that leaders should act as moral models. Through various communication and incentive methods, ethical leaders can shape employees’ moral behavior in the workplace [46]. Therefore, when we further explore empowering leadership, we believe that an empowering leader must be an ethical one first. Only in this way can we ensure that employees exercise their work authority in a responsible and ethical way [47], thus avoiding the occurrence of unethical work behaviors, such as abuses of power and corruption. It is very important. Leaders who empower work on the basis of ethics are more likely to create a cultural atmosphere of trust, respect and fairness in the workplace, which can help employees achieve personal growth and improve their work engagement and retention intentions [48,49]. Therefore, further research on “ethical and empowering” leadership and its positive effects is needed in the future.

### 5.5. Theoretical and Practical Implications

The current research contributes to the existing literature by empirically exploring the role of different job-crafting behaviors in the relationship between empowering leadership and work-related outcomes from a theoretical perspective. Our findings reveal that seeking resources, seeking challenges, and reducing demands play different mediating roles in the association between empowering leadership and work-related outcomes. Future researchers could extend the results of this study by incorporating additional variables to enhance our understanding and generalization of these findings: specifically, whether employees have higher levels of self-efficacy, and to what extent the level of self-efficacy positively affects employees’ job-crafting behavior and its work-related outcome.

From the practical perspective, managers and supervisors can exhibit empowering leadership behavior to initiate and advance employees’ job crafting. To be specific, practitioners can collaborate with managers and supervisors to enhance employees’ autonomy, empowerment, and adaptability in their job roles. This could involve giving employees greater freedom to carry out their duties, providing feedback from diverse sources in various ways, facilitating access to necessary resources, and encouraging the development and utilization of advanced skills whenever employees deem it necessary or desirable. The characteristics of an empowering leader in the context of job crafting include offering support, fostering autonomy, expressing high expectations, and exhibiting a commitment to employee development. Organizations should strive to create more opportunities for leaders to develop these characteristics, such as being able to recognize when employees need job-crafting assistance or want to engage in discussions with leaders to receive support and clarify expectations. Furthermore, job crafting can serve as a valuable tool for organizations and practitioners, enabling employees to proactively adapt to changing environments and situations, thereby mitigating potential performance declines during times of change. To implement and promote this process, our findings can guide practitioners in identifying empowering leadership as an effective strategy for addressing current or anticipated challenges within their organizational contexts [5].

## 6. Conclusions

The current study explored the relationships among empowering leadership, job crafting, work engagement, and in-role performance within the Chinese organizational setting. The results revealed that empowering leadership positively predicts both work engagement and in-role performance, with a stronger predictive capacity for work engagement compared to in-role performance. Furthermore, empowering leadership positively influences employees’ job-crafting behaviors, encompassing seeking resources, seeking challenges, and reducing demands. Compared to reducing demands, empowering leadership more strongly predicts seeking resources and seeking challenges. Seeking resources, seeking challenges, and reducing demands all positively correlate with in-role performance. Additionally, seeking challenges and reducing demands also positively predict work engagement. Seeking resources can better predict in-role performance, and seeking challenges can better predict work engagement. Seeking resources serves as a positive and mediating factor in the linkage between empowering leadership and in-role performance. Similarly, seeking challenges fulfills a positive and mediating role in the relations between empowering leadership, work engagement, and in-role performance. Additionally, reducing demands acts as a positive mediator in the relations among empowering leadership, work engagement, and in-role performance.

## Figures and Tables

**Figure 1 behavsci-14-00451-f001:**
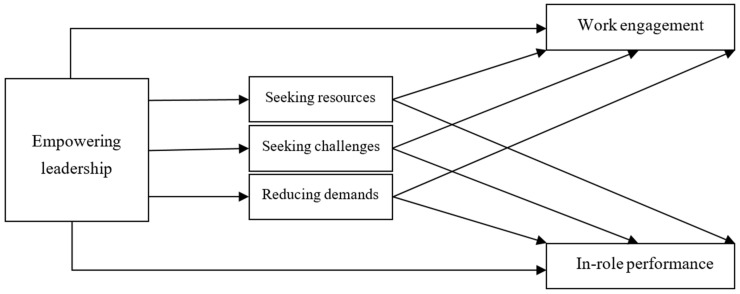
The conceptual model.

**Figure 2 behavsci-14-00451-f002:**
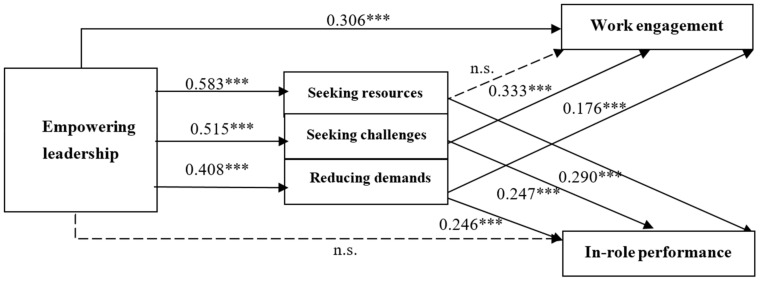
The final model linking empowering leadership to work-related outcomes. Note: The dotted lines indicate hypothesized paths that were not significant. n.s. indicates not significant. *** *p* < 0.001.

**Table 1 behavsci-14-00451-t001:** Means, Standard Deviations, and Intercorrelations of Major Variables (n = 733).

Variables	M	SD	1	2	3	4	5	6	7	8	9	10	11
1. Gender	1.61	0.50	—										
2. Age	3.56	1.52	0.07	—									
3. Education	2.50	0.99	−0.14 **	−0.30 **	—								
4. Tenure	2.63	1.42	0.03	0.65 **	−0.13 **	—							
5. Position	1.50	0.81	−0.19 **	0.20 **	0.01	0.29 **	—						
6. Empowering leadership	3.98	0.70	−0.02	0.01	−0.14 **	0.02	0.12 **	—					
7. Seeking resources	4.36	0.50	−0.01	−0.01	−0.05	−0.01	0.11 **	0.55 **	—				
8. Seeking challenges	3.86	0.68	−0.11 **	0.01	0.03	0.05	0.20 **	0.50 **	0.47 **	—			
9. Reducing demands	3.87	0.68	−0.07	−0.05	−0.05	0.02	0.13 **	0.39 **	0.40 **	0.46 **	—		
10. Work engagement	3.88	0.74	−0.02	0.11 **	−0.21 **	0.08 *	0.19 **	0.53 **	0.44 **	0.57 **	0.45 **	—	
11. In-role performance	4.26	0.55	−0.02	−0.01	−0.02	−0.08 *	0.19 **	0.39 **	0.49 **	0.49 **	0.47 **	0.53 **	—

Note: ** *p* < 0.01; * *p* < 0.05.

**Table 2 behavsci-14-00451-t002:** Summary of Model Fit Indices.

Model	$\chi^{2}$/df	CFI	TLI	AIC	BIC	SRMR	RMSEA (90% CI)
Six factors	3.122	0.950	0.941	32,021.574	32,421.525	0.042	0.043 (0.038, 0.047)
Five factors	6.397	0.854	0.833	32,819.779	33,196.745	0.058	0.072 (0.068, 0.076)
Four factors	5.145	0.891	0.877	32,529.365	32,887.942	0.057	0.062 (0.058, 0.066)
Three factors	8.240	0.798	0.776	33,309.611	33,654.397	0.068	0.083 (0.079, 0.088)
Two factors (a)	8.304	0.709	0.680	34,073.749	34,409.340	0.087	0.100 (0.096, 0.104)
Two factors (b)	10.639	0.730	0.703	33,924.200	34,259.792	0.080	0.096 (0.092, 0.100)
Single factor	13.826	0.640	0.606	34,735.878	35,066.873	0.088	0.111 (0.107, 0.115)

**Table 3 behavsci-14-00451-t003:** Summary of Models Tested and Mplus Fit Statistics (n = 733).

Model	$\chi^{2}$/df	CFI	TLI	AIC	BIC	Adjusted BIC	SRMR	RMSEA (90% CI)
Hypothesized model	4.452	0.942	0.901	12,964.353	13,313.736	13,072.411	0.056	0.069 [0.061, 0.076]
Rival model 1 (The final model)	4.404	0.942	0.903	12,963.447	13,308.233	13,070.083	0.056	0.069 [0.061, 0.076]
Rival model 2	5.002	0.931	0.885	13,015.288	13,360.074	13,121.924	0.068	0.074 [0.067, 0.081]
Rival model 3	4.929	0.932	0.888	13,013.950	13,354.139	13,119.164	0.067	0.073 [0.066, 0.081]

## Data Availability

The data presented in this study are available on request from the corresponding author.

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
