# Peer review of "Crafting a Job among Chinese Employees: The Role of Empowering Leadership and the Links to Work-Related Outcomes"

_behavsci, 2024, doi:10.3390/bs14060451_

Round 1

Reviewer 1 Report

Comments and Suggestions for Authors

The paper provides insightful evidence about the process through which empowering leadership shapes employees' work engagement and in-role performance by facilitating job-crafting behaviors, specifically seeking resources, seeking challenges, and reducing demands. Anyway, there are some issues that still needs improvement and clarification. I encourage the authors to consider the comments given below and revise the paper accordingly in order to enhance the overall quality and completeness of the paper.

1.     In the introduction, it is important to clarify about the knowledge contribution of the paper. How does the paper add important knowledge to fill the gap in literature?

2.     Normally, the conceptual model should not be mentioned in Chapter 1. You may remove “(See Figure 1 The conceptual model)” from the introduction.

3.     For the section header “2.1. The relationship between empowering leadership and work-related outcomes”, it will be better to be more specific on the outcome variables “work engagement” and “in-role performance” instead of mentioning “work-related outcomes” broadly.

4.     Under section “2.3. The relationships among job crafting and work engagement and in-role performance”, the authors just simply mention “regulatory focus theory” without explaining about the key assumption of the theory. This should be briefly explained for readers who may not know about this theory to have some understanding about it.

5.     There are many arguments and sentence about the benefits of empower ring leadership that do not have references to support. For example: Page 3 (employees experience a heightened feeling of freedom or discretion, which facilitate their opportunity perception of what can do in their 124 jobs and how they do). Moreover, many references are also quite old. There are some key papers recently published in high impact-journals that must be incorporated as the references. Please consider adding the papers recommended below for the uncited sentences and to replace the old references:

a.     Using Approach-Inhibition Theory of Power to Explain How Participative Decision Making Enhances Innovative Work Behavior of High Power Distance-Oriented Employees, Journal of Organizational Effectiveness: People and Performance, 10(4), 565-581. https://doi.org/10.1108/JOEPP-10-2022-0304

b.     Reflective and decisive supervision: The role of participative leadership and team climate in joint decision-making. Regulation & Governance, 17: 290-309. https://doi.org/10.1111/rego.12449

6.     Under section “2.3.1. The relationships among seeking resources and work engagement and in-role performance”, the authors mention about the prediction from Job Demands-Resources (JD-R) framework that affects work engagement and superior performance. However, the JD-R model is about stress, not job performance. Using JD-R model to explain work engagement and superior performance directly is misleading. If you still use this theory, please make it clear that work engagement and superior performance could be subsequently achieved when employees cope effectively with stress.

7.     Under section “2.4. The mediating role of employee job crafting”, please have some explanation about the logic to support the mediating effect hypotheses. You can’t just simply have the hypotheses proposed without explanation.

Reviewer 2 Report

Comments and Suggestions for Authors

This was an informative and detailed paper that has the potential to benefit interdisciplinary research in multiple fields. The paper is well written and of an extremely high quality. The findings present a clear contribution to scholarship.

Author Response

Thank you very much for your work. I appreciate your patience.

Reviewer 3 Report

Comments and Suggestions for Authors

The abstract effectively summarizes the content of the article, highlighting the main topics covered and the main results obtained, but it could be more explicit about the type of statistical analysis carried out.

The literature review is rich in references; there is a certain balance between recent and older references, but they could be a little more up to date.

- Clear and correct writing. Just improve aspects, such as separating the words in line 677 "engagement.Seeking".

- Review the bibliographical references. In particular, the fact that the title of the article and the name of the Review/Journal are sometimes separated by commas and sometimes by periods.

Round 2

Reviewer 1 Report

Comments and Suggestions for Authors

The authors did a satisfactory job in this round of revision. The overall quality of the paper is now sufficiently improved. There is no additional comment from my part.